# Increased Prevalence of Atopic Dermatitis in Children Aged 0–3 Years Highly Exposed to Parabens

**DOI:** 10.3390/ijerph182111657

**Published:** 2021-11-06

**Authors:** Johichi Arafune, Hiromasa Tsujiguchi, Akinori Hara, Yukari Shimizu, Daisuke Hori, Thao Thi Thu Nguyen, Fumihiko Suzuki, Toshio Hamagishi, Yohei Yamada, Haruki Nakamura, Takahiro Yoshikawa, Koichiro Hayashi, Aki Shibata, Yuma Fukutomi, Yukihiro Ohya, Kiwako Yamamoto-Hanada, Go Muto, Ryoji Hirota, Tadashi Konoshita, Yasuhiro Kambayashi, Hiroyuki Nakamura

**Affiliations:** 1Department of Environmental and Preventive Medicine, Kanazawa University Graduate School of Medical Sciences, 13-1 Takara-machi, Kanazawa 920-8640, Ishikawa, Japan; j.arafune.home@nifty.com (J.A.); t-hiromasa@med.kanazawa-u.ac.jp (H.T.); ahara@m-kanazawa.jp (A.H.); fumi@dental.email.ne.jp (F.S.); hamagisi@chubu-gu.ac.jp (T.H.); yamada503597@med.kanazawa-u.ac.jp (Y.Y.); haruki_nakamura_kanazawa@yahoo.co.jp (H.N.); yoshikawa-takahiro@snbl.co.jp (T.Y.); k-hayashi@stu.kanazawa-u.ac.jp (K.H.); akintoki1116@gmail.com (A.S.); y-kambayashi@vet.ous.ac.jp (Y.K.); 2Advanced Preventive Medical Center, Kanazawa University, 13-1 Takara-machi, Kanazawa 920-8640, Ishikawa, Japan; 3Department of Nursing, Faculty of Health Sciences, Komatsu University, 14-1 Mukaimotorimachi, Komatsu 923-0961, Ishikawa, Japan; h_zu@me.com; 4Faculty of Medicine, University of Tsukuba, 1-1-1 Tennodai, Tsukuba 305-8575, Ibaraki, Japan; daisuke_hori@md.tsukuba.ac.jp; 5Faculty of Public Health, Haiphong University of Medicine and Pharmacy, Hai Phong 180000, Ngo Quyen, Vietnam; nttthao@hpmu.edu.vn; 6Clinical Research Center for Allergy and Rheumatology, National Hospital Organization Sagamihara National Hospital, 18-1 Sakuradai, Minami-ku, Sagamihara 252-0392, Kanagawa, Japan; fukutomi.yuma.da@mail.hosp.go.jp; 7Allergy Center, National Center for Child Health and Development, 2-10-1 Okura, Setagaya-ku, Tokyo 157-8535, Japan; ohya-y@ncchd.go.jp (Y.O.); yamamoto-k@ncchd.go.jp (K.Y.-H.); 8Department of Hygiene, Kitasato University School of Medicine, 1-15-1 Kitasato, Minami-ku, Sagamihara 252-0374, Kanagawa, Japan; gomuto0123@gmail.com; 9Center for Preventive Medical Science, Chiba University, 1-8-1, Inohana, Chuo-ku, Chiba 260-8670, Japan; 10The Graduate School of Health Science, Matsumoto University, 2095-1 Niimura, Matsumoto 390-1295, Nagano, Japan; ryoji.hirota@t.matsu.ac.jp; 11Third Department of Internal Medicine, University of Fukui Faculty of Medical Sciences, 23-3 Mat-suoka- Shimoaizaki, Eiheiji 910-1193, Fukui, Japan; konosita@u-fukui.ac.jp; 12Department of Public Health, Faculty of Veterinary Medicine, Okayama University of Science, 1-3 Ikoinooka, Imabari 794-8555, Ehime, Japan

**Keywords:** allergy, atopic dermatitis, children, epidemiology, paraben

## Abstract

The prevalence of allergic diseases, such as bronchial asthma, atopic dermatitis, nasal allergies (pollinosis), and food allergies, has been increasing in many countries. The hygiene hypothesis was recently considered from the perspective of exposure to antimicrobial agents and preservatives, such as parabens (CAS number, 94-13-3). It currently remains unclear whether parabens, which are included in many daily consumer products such as cosmetics, shampoos, and personal care products as preservative antimicrobial agents, induce or aggravate allergies. Therefore, the aim of the present study was to examine the relationship between exposure to parabens and the prevalence of allergic diseases in Japanese children. The cross-sectional epidemiology of 236 children aged 0–3 years who underwent health examinations in Shika town in Japan assessed individual exposure to parabens using urinary concentrations of parabens. The results obtained showed that the prevalence of atopic dermatitis was significantly higher in children with high urinary concentrations of parabens than in those with low concentrations (*p* < 0.001). This relationship remained significant after adjustments for confounding factors, such as age, sex, Kaup’s index, and passive smoking (*p* < 0.001). In conclusion, the present results from a population study suggested a relationship between atopic dermatitis and exposure to parabens. A longitudinal study using a larger sample number and a detailed examination of atopic dermatitis, including EASI scores and exposure to parabens, will be necessary.

## 1. Introduction

The prevalence of allergic diseases, such as bronchial asthma, atopic dermatitis, nasal allergies (pollinosis), and food allergies, has been increasing in many countries [1]. The ‘hygiene hypothesis’ has been proposed to explain the increase in allergic diseases due to a modern Western lifestyle and lack of exposure to microorganisms, resulting in a higher incidence of allergies [2,3]. The hygiene hypothesis was recently considered from the perspective of exposure to antimicrobial agents and preservatives, such as parabens and triclosan [4,5,6]. In September 2016, the U.S. FDA (Food and Drug Administration) issued a rule banning the use of triclosan in hand and body washes [7]. However, it currently remains unclear whether parabens, which are included in many daily consumer products such as cosmetics, shampoos, and personal care products as preservative antimicrobial agents [8], induce or aggravate allergies.

Previous studies reported the systemic allergic effects of parabens on allergic diseases, such as asthma [9]. The relationship between exposure to parabens and asthma was examined in the NHANES (National Health and Nutrition Examination Survey), which was conducted as a cross-sectional study on 860 children aged 6–18 years, and the findings obtained showed significantly higher odds of aeroallergen sensitization with increased urinary concentrations of propyl and butyl parabens [6,10]. In contrast, limited information is currently available on the relationship between exposure to parabens and skin lesions; however, an increase in the preservative sensitivity of local skin to parabens has been shown to result in a higher prevalence of allergic contact dermatitis [11,12]. To the best of our knowledge, there is no epidemiology for children younger than three years old to support the relationship between systemic allergic responses and their exposure to parabens. Therefore, the aim of the present study was to examine the relationship between exposure to parabens and the prevalence of allergic diseases in Japanese children aged 0–3 years.

## 2. Materials and Methods

### 2.1. Study Design

Our cross-sectional epidemiology was performed based on the child medical check-up system in Shika town in Japan. Subjects consisted of 236 children who underwent one of the following medical check-ups between January 2017 and March 2019: a 4-month-old medical check-up (77 children), a 1.5-year-old medical check-up (66 children), and a 3-year-old child medical check-up (99 children).

### 2.2. Outcome Assessment

Current food allergies, bronchial asthma, nasal allergies, and atopic dermatitis were assessed by the health check-up doctor based on health records described by the family doctor. Atopic dermatitis was diagnosed using the criteria of the Japanese Dermatological Association [13]. Briefly, patients meeting three basic criteria: having (1) pruritus, (2) the typical morphology and distribution of eczema, and (3) a chronic or chronically relapsing course were considered AD. We also collected data on general demographic characteristics, including age, sex, height, weight, and the passive smoking status. Kaup’s index (kg/m^2^) was calculated from height and weight.

### 2.3. Exposure Assessment

Subjects and their caregivers were examined for all daily commodities used, such as shampoo, toothpaste, body wash, and ointment, to assess their frequency of use and to investigate whether parabens were contained in the daily commodities used in the past 3 days. This information was added to the questionnaire, which is described elsewhere [14]. The study staff checked whether each item included parabens using company information about the product, such as baby lotions, hand or body creams, shampoos, soaps, toothpastes, and ointments. Spot early morning urine samples were collected, and urinary creatinine (Cr) concentrations (mg/dL) were measured by SRL Inc. (Tokyo, Japan). Urinary concentrations of parabens were measured by liquid chromatography–tandem mass spectrometry at Shin Nippon Biomedical Laboratories, Ltd. (Wakayama, Japan), as previously described by Lee-Sarwar et al. [15]. The total amount of parabens was calculated by the sum of µmol/L of methylparaben, ethylparaben, benzylparaben, isobutylparaben, isopropylparaben, butylparaben, and propylparaben. The percentages of each of these parabens among total parabens were as follows: 86.9, 3.97, 0, 0.48, 7.79, 0.42, and 0.40%, respectively. Spearman’s correlation coefficient between total parabens and each paraben were 0.907 (*p* = 0.000), 0.270 (*p* = 0.000), 0.00, 0.547 (*p* = 0.000), 0.018, and 0.270 (*p* = 0.000), respectively. Exposure to parabens was assessed by pmol of the urinary concentration of parabens/mg of Cr. The limits of detection (LOD) were 1 ng/mL for parabens, and the range of detection was 1–100 ng/mL.

### 2.4. Ethics Statement

The present study was approved by the medical Ethics Committee of Kanazawa University (examination number 2184-1). All subjects provided written informed consent.

### 2.5. Statistical Analysis

We used the Student’s *t*-test for continuous variables and the chi-squared test for categorical variables to analyze differences between groups of independent variables (males and females; subjects with and without allergic diseases). We also performed a logistic regression analysis to estimate the independent impact of each variable (age, sex, Kaup’s index, passive smoking, and urinary paraben concentrations) on atopic dermatitis. All hypothesis tests involved two-sided tests, and *p* < 0.05 was considered significant. All analyses were performed using SPSS ver. 24.0.

## 3. Results

### 3.1. Comparisons of Characteristics

Comparisons of characteristics in boys (122 children) and girls (114 children) are shown in Table 1. Age was significantly higher and Kaup’s index was lower in girls than in boys. No significant differences were observed in any allergic diseases, including atopic dermatitis, between boys and girls.

### 3.2. Comparisons of Distribution of Urinary Concentrations of Parabens

The distribution of the urinary concentrations of parabens was compared between children with and without paraben use (Figure 1, left). The concentrations of urinary parabens (pmol/mg Cr) of the maximum, 75th percentile, median, 25th percentile, and minimum were 12,388; 443; 5.75; 0 (limit of quantification); and 0, respectively, in children with paraben use and 6831, 90.5, 0, 0, and 0, respectively, in those without paraben use. The non-parametric statistical method of the Mann–Whitney U test showed that urinary concentrations were not significantly higher but tended to be slightly higher in children with than in those without paraben use (*p* = 0.082). The 75th percentile and median values of the urinary concentrations of parabens in children with paraben use, 443 (pmol/mg Cr, 67.3 μ/g Cr) and 5.75 (pmol/mg Cr, 0.874 μ/g Cr), corresponded to 181.6 nmol/L (27.6 ng/mL) and 2.36 nmol/L (0.358 ng/mL), respectively, after calculations with the median value of the urinary Cr concentration, 401 mg/L.

We also showed the distribution of urinary concentrations of parabens in 31 and 205 children with and without atopic dermatitis, respectively (Figure 1, right). Concentrations of urinary parabens (pmol/mg Cr) for the maximum, 75th percentile, median, 25th percentile, and minimum were 6831, 443, 192, 0, and 0, respectively, in children with atopic dermatitis and 12,388; 83.6; 0; 0; and 0, respectively, in those without atopic dermatitis. The non-parametric statistical method of the Mann–Whitney U test showed that urinary concentrations were significantly higher in children with than in those without atopic dermatitis (*p* = 0.000).

### 3.3. Comparison of the Prevalence of Allergic Diseases among Age Groups

Table 2 shows the prevalence of allergies by health check-up age groups in 236 children, among whom there were 4 with food allergies, 21 with bronchial asthma, 4 with nasal allergies, and 31 with atopic dermatitis. The chi-squared test showed that the prevalence of bronchial asthma was significantly higher in children aged 36 months than in those aged four months.

### 3.4. Comparisons of the Prevalence of Allergic Diseases between High and Low Exposure to Parabens

Table 3 showed a comparison of the prevalence of allergic diseases between children with and without paraben use, which was assessed using the questionnaire. The prevalence of atopic dermatitis was significantly higher in children with than in those without paraben use (*p* = 0.003). Age (*p* < 0.001) and Kaup’s index (*p* = 0.029) were significantly higher in children with than in those without paraben use.

When exposure to parabens was assessed based on urinary concentrations, children were divided into two groups: those with high urinary concentrations of parabens of more than 100 (pmol/Cr, *N* = 61) and those with low concentrations of less than 100 (pmol/Cr, *N* = 175). The prevalence of atopic dermatitis was significantly higher in children with high urinary concentrations of parabens than in those with low concentrations (*p* < 0.001, Table 4). Eleven boys (31.4%) and seven girls (26.9%) had atopic dermatitis with high urinary concentrations of parabens. The prevalence of atopic dermatitis was significantly lower in children with low urinary concentrations of parabens (five boys (5.7%, *p* < 0.001) and eight girls (9.1%, *p* = 0.026)). Regarding food and nasal allergies, the very small number of cases examined did not allow for a statistical analysis of differences in their prevalence between the two groups.

Since confounding factors, including Kaup’s index, differed between the two groups in Table 4, we performed a multiple logistic analysis of atopic dermatitis with variables, including urinary concentrations of parabens and confounding factors. The results obtained showed that the relationship between urinary paraben concentrations and atopic dermatitis in children remained significant after adjustments for confounding factors, such as age, sex, Kaup’s index, and passive smoking (*p* < 0.001, Table 5).

## 4. Discussion

Individual exposure to parabens in the living environment may be assessed by questionnaires on the daily use of consumer products, such as cosmetics, shampoos, and personal care products, but more objectively by an examination of urinary concentrations of parabens [16,17]. Previous studies [16,18,19] reported that urinary concentrations of parabens increased in a dose-dependent manner with a higher frequency of use of several cosmetic products. In the present study, urinary concentrations of parabens tended to be slightly higher in children with than in those without paraben use. The present results on not only paraben use but also urinary paraben concentrations in children demonstrated that the prevalence of atopic dermatitis was significantly higher in children more highly exposed to parabens than in those less exposed. The NHANES [4] reported a positive relationship between allergic sensitization and urinary concentrations of parabens. Berger et al. [20] showed that urinary methylparaben was associated with lower Th1%. However, limited information is currently available on the relationship between paraben exposure and atopic dermatitis. Overgaard et al. [21] reported that children aged 4–9 years with atopic dermatitis and using emollients had elevated urinary concentrations of parabens. A preliminary study also showed that urinary concentrations of parabens were associated with current atopic dermatitis in children younger than 15 years [9]. In a review by Jackson-Browne [22], a small number of studies suggested that paraben exposure was related to the risk of eczema in children, possibly through changes in microbiome diversity and immune function from antimicrobial exposure. Thürmann and his colleagues [23], who examined prenatal exposure to parabens using maternal urine samples in the German mother–child study LINA, recently demonstrated that exposure to ethylparaben and n-butylparaben was associated with an increased risk of the development of very early onset AD. In contrast to the findings of studies that focused on older children in hospital or prenatal exposure, the present results on atopic dermatitis in children younger than three years old are considered of importance due to epidemiological implications and the pathogenesis of atopic dermatitis. Our study using a general population enabled the assessment of paraben exposure as one of the environmental factors potentially impacting the etiology of disease. Furthermore, using our epidemiological design, we eliminated the effects of being obese and overweight. Based on the assumption that exposure to parabens affects anthropometric parameters [24] as well as obesity and metabolic syndrome [25], our multiple logistic analysis with confounding factors, including Kaup’s index, suggested that parabens are a risk factor for the development of atopic dermatitis in children.

Regarding the pathogenesis of atopic dermatitis, an increase in the preservative sensitivity of local skin to parabens has been shown to result in a higher prevalence of allergic contact dermatitis [11,12]. On the other hand, Lee-Sarwar et al. [15] reported an inverse relationship between paraben exposure and allergic sensitization. Taken together with the relationship between parabens, the effects of biocides [8], and allergies, the hygiene hypothesis is considered the most likely explanation for the relationship between atopic dermatitis and parabens, which are antimicrobial agents, resulting in the predominance of Th2. Since Lee et al. [26] demonstrated that propylparaben exposure was associated with the Eczema Area and Severity Index (EASI score), partially via metabolomic changes related to oxidative stress, atopic dermatitis in the present study may have been induced by parabens through a different pathway to the predominance of Th2.

The present results also showed that the increased prevalence of atopic dermatitis in children aged 0–3 years was important because exposure to parabens in their early lives may have resulted in the early onset of atopic dermatitis [1,4,5,6]. Regarding the pathogenesis of the early onset of atopic dermatitis, the predominance of Th2 accompanying exposure to parabens is considered to lead to the onset of atopic dermatitis. However, Quiros-Alcala et al. [27] reported that inconsistent findings on the effects of paraben exposure on allergic sensitization, including the incidence of bronchial asthma, were due to sex differences. The present results showed no significant differences in the involvement of atopic dermatitis and paraben exposure between sexes. Therefore, the effects of parabens on allergic diseases do not appear to be due to sex differences. Furthermore, the RHINESSA study in Bergen, Norway [19], on 496 adults and 90 adolescents reported contrasting findings to the present results, namely, urinary paraben concentrations were lower in subjects with current eczema. The discrepancies between the RHINESSA study and the present results may be attributed to differences in the ages of subjects; however, it is important to note that the urinary concentration of methylparaben, which accounts for most urinary parabens [28,29], in the RHINESSA study was 5.63 ng/mL in adolescents, which was markedly lower than 55.1 μg/g Cr in 3–5 year old children examined by the Korean National Environmental Health Survey [30]. A comparative study of urinary paraben concentrations between Asian and Western countries performed by Honda et al. [28] revealed higher concentrations in Asian countries, including Korea, Japan, and India. The present study also calculated the 75th percentile value of the urinary concentrations of parabens in children with paraben use to be 67.3 μ/g Cr (27.6 ng/mL), which was higher than that in the RHINESSA study [19]. In the RHINESSA study [19], the regulation of hygiene points of view forced the industry in Western countries to replace parabens with alternative preservatives, which decreased urinary concentrations of parabens. This speculation may account for the inverse relationship observed between the prevalence of eczema and paraben concentrations in the RHINESSA study [19]. One of the etiologies of atopic dermatitis in children in countries such as Asian countries, in which children frequently encounter parabens in their daily lives, is still continuous exposure to parabens.

There were several limitations that need to be addressed. The present study used a cross-sectional design that was unable to assess the causal relationship between disease and exposure to parabens. The more frequent use of ointments in children with atopic dermatitis may increase exposure to parabens, subsequently leading to higher urinary concentrations of parabens [30]. We examined exposure to parabens in a single urine sample, not at several time points. Furthermore, the small number of children examined prevented analyses of allergic diseases other than atopic dermatitis. In the future, a longitudinal study using a larger sample number and a detailed examination of atopic dermatitis, including EASI scores and exposure to parabens, will be necessary.

## 5. Conclusions

The cross-sectional epidemiology of 236 children aged 0–3 years in Shika town in Japan showed that the prevalence of atopic dermatitis was significantly higher in children exposed to parabens, which was assessed using urine samples. This relationship remained significant after adjustments for confounding factors, such as age, sex, Kaup’s index, and passive smoking. The present results obtained from a population study suggested a relationship between atopic dermatitis and exposure to parabens. A longitudinal study using a larger sample number and a detailed examination of atopic dermatitis, including EASI scores and exposure to parabens, will be necessary.

## Figures and Tables

**Figure 1 ijerph-18-11657-f001:**
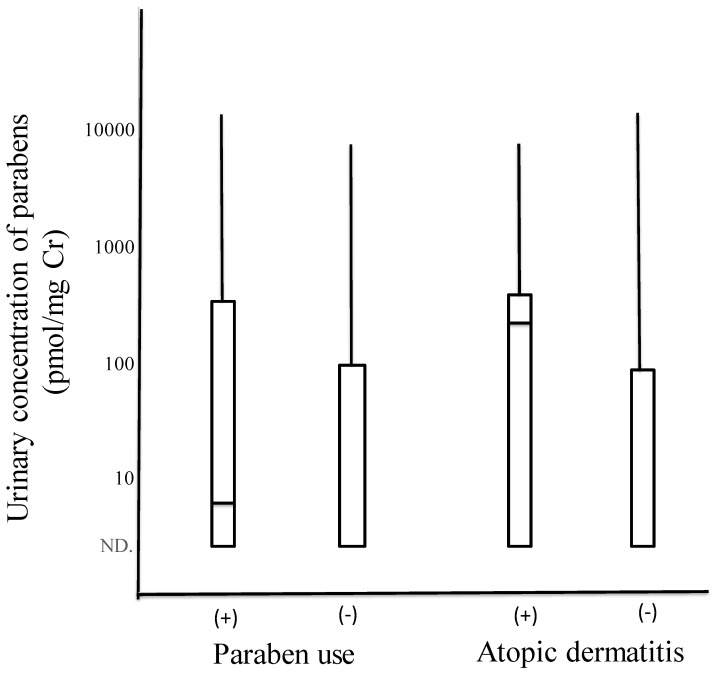
Comparisons of the distribution of urinary concentrations of parabens between children with and without paraben use (**left**) and between children with and without atopic dermatitis (**right**). Each boxplot represents maximum, 75th percentile, median, 25th percentile, and minimum.

**Table 1 ijerph-18-11657-t001:** Characteristics of children in the present study according to sex.

	Boy (*N* = 122)	Girl (*N* = 114)
Age (month (mean ± SD))	20.42 ± 16.66	25.72 ± 17.36 *
Height (cm (mean ± SD))	78.81 ± 14.44	82.08 ± 15.25
Weight (kg (mean ± SD))	10.61 ± 3.67	11.36 ± 3.75
Kaup’s index (kg/m^2^ (mean ± SD))	16.75 ± 1.68	16.32 ± 1.36 *
Number with passive smoking (percentage)	41 (34.2%)	34 (29.8%)
Food allergies (number (prevalence))	4 (3.3%)	7 (6.1%)
Bronchial asthma (number (prevalence))	10 (8.2%)	11 (9.6%)
Nasal allergies (number (prevalence))	1 (0.8%)	3 (2.6%)
Atopic dermatitis (number (prevalence))	16 (13.1%)	15 (13.2%)
Total allergies (number (prevalence))	27 (22.1%)	31 (27.2%)
Number using parabens (percentage)	36 (29.5%)	46 (40.4%)
Number with high urinary concentrations of parabens (percentage)	35 (28.7%)	26 (22.8%)

* The Student’s *t*-test showed a significant difference from the value for boys; *p* < 0.05.

**Table 2 ijerph-18-11657-t002:** Prevalence of allergic diseases and other factors according to age groups.

Age Group (Number)	4 Months (*N* = 77)	18 Months (*N* = 60)	36 Months (*N* = 99)	*p* Value
Number of boys (percentage)	45 (58.4%)	36 (60.0%)	41 (41.4%)	0.026
Age (month (mean ± SD))	3.12 ± 0.32	17.25 ± 3.90	41.9 ± 2.48	0.000
Height (cm (mean ± SD))	62.11 ± 2.37	79.08 ± 2.87	95.59 ± 3.87	0.000
Weight (kg (mean ± SD))	6.66 ± 0.82	10.53 ± 0.99	14.58 ± 1.96	0.000
Kaup’s index (kg/m^2^ (mean ± SD))	17.23 ± 1.57	16.9 ± 1.19	15.8 ± 1.40	0.000
Number with passive smoking (percentage)	22 (28.6%)	18 (30.0%)	35 (36.1%)	0.534
Food allergies (number (prevalence))	0 (0.0%)	0 (0.0%)	4 (4.0%)	-
Bronchial asthma (number (prevalence))	0 (0.0%)	6 (10.0%)	15 (15.2%)	0.002
Nasal allergies (number (prevalence))	0 (0.0%)	0 (0.0%)	4 (4.0%)	-
Atopic dermatitis (number (prevalence))	15 (19.5%)	4 (6.7%)	12 (12.1%)	0.082
Number using parabens (percentage)	11 (14.3%)	20 (33.3%)	51 (51.5%)	0.000
Number with high urinary concentrations of parabens (percentage)	23 (29.9%)	17 (28.3%)	21 (21.2%)	0.380

An analysis of variance was used to compare age, height, weight, and Kaup’s index among age groups. The chi-squared test was used to compare the proportions of boys, passive smoking, food allergies, bronchial asthma, nasal allergies, atopic dermatitis, paraben use, and high urinary concentrations of parabens.

**Table 3 ijerph-18-11657-t003:** Comparisons of prevalence of allergic diseases between children with and without paraben use.

Paraben Use (Number)	(Paraben Use +) (*N* = 82)	(Paraben Use −) (*N* = 154)	*p* Value
Number of boys (percentage)	36 (43.9%)	86 (55.8%)	0.082
Age (month (mean ± SD))	30.44 ± 15.50	19.01 ± 16.73	0.000
Kaup’s index (kg/m^2^ (mean ± SD))	16.24 ± 1.47	16.71 ± 1.57	0.029
Number with passive smoking (percentage)	23 (28.4%)	52 (33.7%)	0.385
Food allergies (number (prevalence))	2 (2.4%)	2 (1.3%)	-
Bronchial asthma (number (prevalence))	10 (12.2%)	11 (7.1%)	0.194
Nasal allergies (number (prevalence))	1 (1.2%)	3 (1.9%)	-
Atopic dermatitis (number (prevalence))	18 (22.0%)	13 (8.4%)	0.003
Number with high urinary concentrations of parabens (percentage)	27 (32.9%)	34 (22.1%)	0.082

**Table 4 ijerph-18-11657-t004:** Comparisons of prevalence of allergic diseases between children with high and low urinary concentrations of parabens.

Urinary Concentrations of Parabens	High (Higher than 100 (pmol/Cr, *N* = 61))	Low (Lower than 100 (pmol/Cr, *N* = 175))	*p* Value
Number of boys (percentage)	35 (57.4%)	87 (49.7%)	0.304
Age (month (mean ± SD))	19.79 ± 17.45	24.09 ± 16.98	0.092
Kaup’s index (kg/m^2^ (mean ± SD))	16.72 ± 1.61	16.48 ± 1.52	0.308
Number with passive smoking (percentage)	17 (28.3%)	58 (33.3%)	0.476
Food allergies (number (prevalence))	0 (0.0%)	4 (2.3%)	-
Bronchial asthma (number (prevalence))	6 (9.8%)	15 (8.6%)	0.765
Nasal allergies (number (prevalence))	0 (0.0%)	4 (2.3%)	-
Atopic dermatitis (number (prevalence))	18 (29.5%)	13 (7.4%)	0.000
Number using parabens (percentage)	27 (44.3%)	55 (31.4%)	0.082

**Table 5 ijerph-18-11657-t005:** Multiple logistic analysis of atopic dermatitis with variables including the urinary concentration of parabens.

	β (SE)	Exp (β) (95% Confidence Interval)	*p* Value
Boy	−0.154 (0.411)	0.857 (0.383–1.919)	0.708
Age	0.007 (0.013)	1.007 (0.981–1.033)	0.602
Kaup’s index	0.013 (0.143)	1.013 (0.765–1.341)	0.928
Passive smoking	−0.067 (0.445)	0.935 (0.390–2.238)	0.880
Group with high urinary concentrations	1.609 (0.407)	4.995 (2.248–11.099)	0.000

## Data Availability

The data presented in this study are available from the corresponding author upon request. The data are not publicly available due to policies of the ethics committee.

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
