# Peer review of "Increased Prevalence of Atopic Dermatitis in Children Aged 0–3 Years Highly Exposed to Parabens"

_ijerph, 2021, doi:10.3390/ijerph182111657_

Round 1
Reviewer 1 Report
In this study Arafune et al investigated the use and excretion of parabens regarding the prevalence of allergic disease in Japanese children aged 0-3 years. The authors declared that the use of parabens was significantly associated with the prevalence of AD and suggested that the use of parabens promote the development of AD. The manuscript is well written and well organized. However I have several questions.
- It is not clear how the authors assessed the “use of parabens”. Could you provide a list with the shampoos, body wash and ointments, including containing parabens? Children with AD are normally instructed to apply ointments daily as they reduce e.g. the use of corticosteroids. How can you exclude that the more frequent use of ointments in AD patients subsequently leads to a higher use of parabens and not the other way around?
- Did you measure different parabene species (meth[1]ylparaben (MeP), ethylparaben (EtP), isopropylparaben (iPrP), n[1]propylparaben (nPrP), sec-butylparaben (sBuP), isobutylparaben (iBuP), n-butylparaben (nBuP),benzylparaben(BzP)? And is there an association for each paraben in regard of AD onset?
- Figure 1. Values should be shown as mean +- SEM (if normally distributed) or median.
Shouldn’t the paraben use reflect the urinary concentration, like it has been shown by Overgaard et al? (Figure 1 and Table 1)
- What is the message behind Table 2?
- Since this is the central message a graph including the urinary paraben concentration between children with AD and no AD should be included.
- Is there an association between EASI score and urinary paraben levels?
- On what basis did the authors set the cut off of 100pmol/Cr parabene concentration?
- Regarding tables: Why do the authors switch between percentage and prevalence? Pls. include also percentage behind food allergies, asthma and so on.
- Recently published Paper, like Thürmann et al. 2021 (Allergy) or Lee et al. 2021 (Sci Rep) should be discussed
Author Response
First of all, I appreciate the time and effort you have dedicated to providing insightful feedback on ways to strengthen our paper. I have incorporated changes that reflect the detailed suggestions you have graciously provided.
1. Thank you very much for your important suggestions in which you have commented that the more frequent use of ointments in AD patients leads to a higher use of parabens. Following the comments, I have added in the limitations in the section of Discussion that “Especially, the more frequent use of ointments in children with atopic dermatitis patients might increase use of parabens, subsequently leading to the higher urinary level of parabens [30].”
2. Thank you very much for your important comments. In the revised manuscript, I have added some data about different parabene species in 2.3. Exposure assessment in the section of Material and Methods in the following; “The total amount of parabens was calculated by the sum of µmol/L of methylparaben, ethylparaben, benzylparaben, isobutylparaben, isopropylparaben, butylparaben and propyparaben. The percentage of each paraben for total parabens were 86.9 %, 3.97 %, 0 %, 0.48 %, 7.79 %, 0.42 %, and 0.40 %, respectively. The Spearman’s correlation coefficient between total parabens and each paraben were 0.907 (p=0.000), .0.270 (p=0.000), 0.00, 0.547 (p=0.000), 0.018, and 0.270 (p=0.000), respectively.”
3. I thank you for your comments regarding the interpretation of the Fig. 1. In the 3.2.in the section of Results, I have described in the followings; “The distribution of urinary concentrations of parabens was compared between children using and not using parabens (Fig. 1 (left)). Concentrations of urinary parabens (pmol/mg Cr) of maximum, 75 percentile, median, 25 percentile, and minimum were 12388, 443, 5.75, 0 (limit of quantification), 0 in children with paraben use, and 6831, 90.5, 0, 0, 0 without paraben use, respectively. The non-parametric statistical method of the Mann-Whitney U-test showed that urinary concentrations were slightly higher in children with than without paraben use (p=0.082).”
Therefore, I consider that the paraben use reflects almost the urinary concentration.
- I have shown in the Table 2 that the prevalence of atopic dermatitis including other allergic diseases such as food allergies, bronchial asthma, nasal allergies may change with the age of children, in the same pattern as the average Japanese children.
5. I thank you very much for your important comments. Following your comments, I have added that additional graph in the right of the Fig.1, which shows the urinary paraben concentration between children with AD and no AD.
6. I thank you very much for your important comments. In the last of the section of Discussion, “In the future, a longitudinal study using larger samples and the detailed examination for atopic dermatitis including EASI score and exposure to parabens will be necessary.”
7. The 75 percentile in the urinary concentration of paraben in total 236 subjects almost corresponds to the level of 100 pmol/mg Cr. If the median were used as the cut- off point, the level would be nearly almost zero pmol/mg Cr
8. I have used the term “prevalence” only in the case of diseases. In other case, I have used ”percentage”.
9. Following your important information, I have added in the discussion section the literatures of Thürmann et al. 2021 (Allergy) or Lee et al. 2021 (Sci Rep) and expanded and deepened the discussion.
Reviewer 2 Report
v
It is an interesting paper on the frequency of atopic dermatitis and other allergic diseases in children aged 0-3 years in the context of parabens exposure.
The study group consisted of 236 patients. The study was based on a survey (paraben exposure) and measurement of paraben concentration in the urine.
The manuscript does meet the review criteria.
I have major concerns :
The abstract is immature and needs to be completely rebuilt – please add results, change conclusions as in the main text.
Introduction – other hypotheses for the development of allergies should be added.
It would be good to write what is the frequency of allergic diseases, including AD in children in Japan in this age group (0-3y).
In section 2.1, please describe that this was a cross-sectional study, how many times was a particular child examined? It would be good to add a graphical description of the study designed
Section 2.2 what were the criteria for the diagnosis of asthma, allergic rhinitis, atopic dermatitis. Were the degrees of severity of allergic diseases determined – what were the criteria?
Author Response
- Following the suggestion regarding the abstract, I have changed in the result and conclusion corresponding the main text.
- Thank you very much for your suggestion regarding other hypotheses for the development of allergies. Following your suggestion, I have added in the section of Discussion in the following;
“Since Lee et al [26] have demonstrated that propyl-paraben exposure was associated with Eczema Area and Severity Index (EASI score), partially via metabolomic changes related with oxidative stress, atopic dermatitis seen in this study might be induced by parabens through the pathway different from predominance of Th2.”
- To clarify the study design, following your suggestion, I have changed 2.1 Study design to the following description; “Our cross-sectional epidemiology was performed on the base of the child medical checkup system in Shika town in Japan. Subjects consisted of 236 children who underwent one of the following medical check-up; a 4-month medical check-up (77 children), a 1.5-year-old medical check-up (66 children), a 3-year-old child medical check-up (99 children) between January 2017 and March 2019.”
- In Section 2.2, I have added the criteria of the atopic dermatitis in the following; “Atopic dermatitis was diagnosed by the criteria of Japanese Dermatological Association [13]. Briefly, patients meeting three basic criteria, 1) pruritus, 2) typical morphology and distribution of the eczema, and 3) chronic or chronically relapsing course, were regarded as having AD.”
Furthermore, as we have not analyzed the association between EASI score and the urinary paraben concentration, I have added in the last sentence of the discussion; “In the future, a longitudinal study using larger samples and the detailed examination for atopic dermatitis including EASI score and exposure to parabens will be necessary.”
Reviewer 3 Report
The study results are well presented.
The limitations have been adequately reported by the Authors.
However, among limitations, I would add that questionnaires + single urine sample are not the ideal way to have a clear picture of the paraben exposure. On the other hand, I understand it is difficult to find alternative methods. I would mention that analyzing longitudinal urine samples from the enrolled subjects (confirming high exposure over time) would strengthen the work.
Considering these limitations, I think that the final sentence in the conclusion is too strong.
With this cross-sectional study, the Authors report/confirm an interesting association, atopy and parabens exposure, but it is not possible to define causality.
Author Response
I thank you very much for your significant suggestion. Following your suggestion, I have changed the last sentence of the discussion in the following; There were several limitations that need to be addressed. The present study used a cross-sectional design, which could not determine a causal relationship between disease and exposure to parabens. Especially, the more frequent use of ointments in children with atopic dermatitis patients might increase use of parabens, subsequently leading to the higher urinary level of parabens [30]. We assessed the exposure to parabens in single urine sample, not in several time points. Furthermore, the small number of children examined prevented analyses of allergic diseases other than atopic dermatitis. In the future, a longitudinal study using larger samples and the detailed examination for atopic dermatitis including EASI score and exposure to parabens will be necessary.”
Round 2
Reviewer 1 Report
The manuscript improved due to the revision process. However some questions have not yet been sufficiently addressed.
1. Could you provide a list with most frequently used shampoos, body wash and ointments, containing parabens in your study?
2. Regarding Figure 1. ”. It is not clear from the plot that the median in the groups (paraben +/-) is 5.75 and 0 respectively. Could you please choose a different presentation style, e.g. box plots?
And I could not find the information (median) regarding AD (+/-). It is not described in the text.
Author Response
Comments and Suggestions for Authors (Reviewer 1)
The manuscript improved due to the revision process. However some questions have not yet been sufficiently addressed.
1. Could you provide a list with most frequently used shampoos, body wash and ointments, containing parabens in your study?
2. Regarding Figure 1. ”. It is not clear from the plot that the median in the groups (paraben +/-) is 5.75 and 0 respectively. Could you please choose a different presentation style, e.g. box plots?
And I could not find the information (median) regarding AD (+/-). It is not described in the text.
Responses (Reviewer 1)
First of all, I appreciate the time and effort you have dedicated to providing insightful feedback on ways to strengthen our paper. I have incorporated changes that reflect the detailed suggestions you have graciously provided.
1. Thank you very much for your important suggestions. I have prepared a list with most frequently used shampoos, body wash and ointments, containing parabens in your study, and uploaded the file entitled “Paraben-containing products.docx”.
- Thank you very much for your important comments. Following your comments, In the Figure1, I have added boxplots according to each group. In the section of “3.2 Comparisons of distribution of urinary concentrations of parabens” (result section), I have described in the following: “Concentrations of urinary parabens (pmol/mg Cr) for the maximum, 75th percentile, median, 25th percentile, and minimum were 6831, 443, 192, 0, and 0, respectively, in children with atopic dermatitis, and 12388, 83.6, 0, 0, and 0, respectively, in those without atopic dermatitis.”
Reviewer 2 Report
Thank you for the opportunity of review your manuscript.
I have some observations.
- In abstract and main text:
methods: please add, informations about :
survey assessed Individual exposure to parabens; and assessment of parabens concentrations in Urinary - not only check-up and informations from medical records.
line 47: add p-value
conclusion :
in limitation the Authors wrote:
„ The present study used a cross-sectional design, which could not determine a causal relationship between disease and exposure to parabens.”
But you wrote in conclusion :
„In conclusion, the present results from a population study revealed a relationship between atopic dermatitis and exposure to parabens.”
Please explain
Line 80 chapter 2; line 81 – sub-chapter 3.1? –please correct
line 125 The Authors wrote: „p<0.05 was considered to be significant” but in line 141 wrote : „urinary concentrations were slightly higher in children with than without paraben use (p=0.082). „
Please explain this.
The children in the paraben-use group were statistically older than children without paraben use. Please comment on this in the text.
Author Response
Comments and Suggestions for Authors (Reviewer 2)
Thank you for the opportunity of review your manuscript. I have some observations.
- In abstract and main text:
methods: please add, informations about survey assessed Individual exposure to parabens; and assessment of parabens concentrations in Urinary - not only check-up and informations from medical records.
line 47. add p-value
- conclusion :
in limitation the Authors wrote:
The present study used a cross-sectional design, which could not determine a causal relationship between disease and exposure to parabens.”
But you wrote in conclusion :
„In conclusion, the present results from a population study revealed a relationship between atopic dermatitis and exposure to parabens.”
Please explain
- Line 80. chapter 2; line 81 – sub-chapter 3.1? –please correct
- line 125 The Authors wrote: „p<0.05 was considered to be significant” but in line 141 wrote: „urinary concentrations were slightly higher in children with than without paraben use (p=0.082). „
Please explain this.
- The children in the paraben-use group were statistically older than children without paraben use. Please comment on this in the text.
Responses (Reviewer 2)
First of all, I appreciate the time and effort you have dedicated to providing insightful feedback on ways to strengthen our paper. I have incorporated changes that reflect the detailed suggestions you have graciously provided.
- I thank you very much for your significant comments. Following your comments, I have added in the section of abstract in the following: “The cross-sectional epidemiology of 236 children aged 0-3 years who underwent health examinations in Shika town in Japan assessed individual exposure to parabens using urinary concentrations of parabens.”
I have added “p value” in the section of abstract and results.
- Following your kind comments, in the section of abstract and conclusion I have changed in the following:
In the abstract: “In conclusion, the present results from a population study suggested a relationship between atopic dermatitis and exposure to parabens.”
In the conclusion: “The present results obtained from a population study suggested a relationship between atopic dermatitis and exposure to parabens.”
- Sub-chapter 3 has changed to subchapter 2.
- Following your kind comments, in the section of results I have changed in the following: “The non-parametric statistical method of the Mann-Whitney U-test showed that urinary concentrations were not significantly, but tended to be slightly higher in children with than in those without paraben use (p=0.082).”
- In the section of results, I have added the results regarding the differences in the ages between the children with and without paraben use in the following:
“Age was significantly higher and Kaup’s index was lower in girls than in boys.”
I suppose the reasons why the children in the paraben-use group were statistically older than children without paraben use is due to the facts that many mother tend to take care of baby using organic baby lotion without parabens during baby period.